# Building systems for preparedness: Global scoping studies on institutional governance and National Public Health Agencies

Sileshi Demelash Sasie[1]*, Fantu Mamo Aragaw[2], Tigist Ali Gebeyehu[1], Sintayehu Abdella Shikur[1], Lensa Fekadu[1], Neima Zeynu Ali[1], Zenebech Mamo Argaw[3]

**1** Ethiopian Public Health Institute, Addis Ababa, Ethiopia, **2** College of Medicine and Health Sciences, University of Gondar, Gondar, Ethiopia, **3** Department of Public Health, College of Medicine and Health Sciences, Wollo University, Dessie, Ethiopia

\* sileyeshi21@gmail.com

## Abstract

Public health emergencies remain a persistent threat to global health security, with the COVID-19 pandemic exposing critical weaknesses even in advanced health systems. National Public Health Agencies (NPHAs), particularly National Public Health Institutes (NPHIs), have emerged as central actors in coordinating preparedness and response functions. However, institutional maturity, financing, and subnational integration remain uneven, especially in low- and middle-income countries. This scoping review consolidates evidence on governance, institutional arrangements, workforce development, financing, and cross-cutting determinants shaping public health emergency preparedness and response. A scoping review of literature published between 2000 and 2025 was conducted following the Arksey and O'Malley framework and reported according to PRISMA-ScR standards. Systematic searches were performed in PubMed, Scopus, Web of Science, and relevant WHO repositories. Eligible studies were screened and charted using a standardized data extraction template, and findings were synthesized using an inductive thematic approach. A total of 4,163 records were identified, 538 duplicates removed, 3,625 records screened, 98 full texts assessed, and 60 studies included. Evidence was organized into seven domains: institutional landscape, governance and autonomy, preparedness functions, workforce, financing, subnational presence, and cross-cutting enablers. National public health institutes were widely established but frequently operated with unclear mandates and limited subnational implementation. Core functions such as surveillance, laboratory systems, risk communication, and emergency operations were inconsistently institutionalized, with recurrent constraints in financing, workforce capacity, and digital interoperability across regions. Public health emergency preparedness and response systems function most effectively when embedded within stable governance, financing, and accountability frameworks. Strengthening statutory

**Data availability statement:** All data analyzed in this study were obtained from published and grey literature sources. These are included within the manuscript and its supplementary materials.

**Funding:** The authors received no specific funding for this work.

**Competing interests:** The authors have declared that no competing interests exist.

authority, predictable domestic investment, workforce sustainability, and interoperable information systems is critical to translating technical capacity into reliable operational performance. Durable, institutionally anchored preparedness is essential for moving from reactive crisis management to sustained and equitable national readiness.

## Background

At the global level, public health emergencies remain among the most pressing threats to health security, economic stability, and social resilience. The COVID-19 pandemic, which has caused more than 700 million confirmed cases and over 7 million deaths worldwide, exemplified how even the most advanced health systems can be quickly overwhelmed [1,2]. Beyond COVID-19, recurring epidemics of influenza, cholera, Ebola, and vector-borne diseases, alongside climate-driven disasters and humanitarian crises, continue to demand rapid, coordinated, and sustained responses [3,4]. High-income countries with established infrastructures struggled to maintain essential services, sustain workforce capacity, and ensure multisectoral coordination [5,6]. These experiences confirm that no health system is immune to disruption, and that gaps in preparedness whether in governance, surveillance, or supply chains translate into preventable morbidity, mortality, and social disruption [7–11].

In low- and middle-income countries (LMICs), the magnitude of the problem is even more severe. Chronic underinvestment in health infrastructure, fragile surveillance and laboratory networks, and limited surge financing mechanisms heighten vulnerability to public health threats [3,12]. In sub-Saharan Africa, fewer than one-third of countries operate fully functional Public Health Emergency Operations Centres (PHEOCs), and health workforce surge capacity remains inadequate [12,13]. Ethiopia illustrates these systemic challenges: despite progress in building a decentralized public health system, recurrent cholera and malaria outbreaks, climate-induced shocks, and conflict-related displacement continue to strain national and sub-national preparedness [3,12]. National Public Health Institutes (NPHIs), including the Ethiopian Public Health Institute (EPHI), often contend with fragmented mandates, weak autonomy, and unpredictable funding flows, which undermine their ability to mount timely and coordinated responses [2,3]. The consequence is delayed outbreak detection, insufficient cross-sectoral collaboration, and greater human and economic costs at the community level [2,3]. For consistency, this review uses the term National Public Health Institute (NPHI) as an overarching designation for national-level public health bodies responsible for surveillance, laboratory coordination, and public health emergency preparedness and response. In the global literature, the terms National Public Health Agency (NPHA) and National Public Health Institute (NPHI) are often used interchangeably to describe institutions with similar mandates. Where included studies used either term, both were considered functionally equivalent for the purposes of this synthesis.

Recognizing these challenges, a range of global and regional initiatives have been established to strengthen preparedness and response. The International Health

Regulations (IHR 2005) provide a legally binding framework for 196 States Parties, supported by tools such as the Joint External Evaluation (JEE), the State Party Self-Assessment Annual Reporting (SPAR), and the After-Action Review (AAR) methodology [14,15]. WHO has also advanced the Emergency Response Framework (ERF), the Health Emergency and Disaster Risk Management (Health-EDRM) framework, and technical guidance for maintaining essential services and managing infodemics [16]. Most recently, WHO launched the Global Health Emergency Corps (GHEC) to foster a world-wide network of trained, interoperable experts and institutions ready to deploy during crises, complementing mechanisms such as the Global Outbreak Alert and Response Network (GOARN) and Emergency Medical Teams (EMTs) [16,17]. Regionally, Africa CDC has championed innovations such as the Event-Based Surveillance Framework, the Saving Lives and Livelihoods (SLL) initiative, and technical guidelines for biosafety, biosecurity, and PHEOCs [18,19]. At national levels, many countries have adopted workforce training initiatives including Field Epidemiology Training Programmes (FETPs) and emergency preparedness plans to enhance institutional and community resilience [18,20].

Despite notable advances, critical shortcomings persist in existing efforts. Many initiatives remain fragmented across institutions, narrowly focused on disease-specific preparedness, or confined to single-country case studies [3,21]. Comparative assessments of how national public health institutes (NPHIs) and ministries of health deliver core functions such as governance, workforce development, laboratory readiness, and risk communication remain limited [21,22]. Where evidence does exist, it is often inconsistent or poorly standardized, hindering cross-country benchmarking and shared learning [8,22]. Moreover, prevailing tools tend to emphasize technical capacities while giving insufficient attention to structural and institutional determinants, including governance arrangements, sustainable financing, and scientific independence factors that ultimately shape system performance during real emergencies [21,23]. These gaps are particularly consequential for low- and middle-income countries, where the translation of global frameworks into effective local implementation is often weak or uneven [3,24]. Although substantial literature describes individual components of preparedness, less attention has been paid to how these components interact as part of an integrated institutional system. This review therefore advances a central argument: preparedness effectiveness depends less on the mere presence of institutions than on their integration within legally mandated, autonomous, and sustainably financed governance systems with empowered subnational implementation. For consistency, this manuscript uses the term public health emergency preparedness and response (PHEPR) to refer to the full range of capacities required to prevent, detect, prepare for, and respond to public health emergencies. Related terms such as PHEP and EPR appear in the literature but are not used interchangeably in this review.

This scoping study seeks to address these gaps by systematically reviewing and synthesizing global and regional evidence on the role of NPHIs and ministries of health in public health emergency preparedness and response [3,24]. Using the WHO–NPHI global mapping instrument as an organizing framework, the review examines how country's structure, resource, and assign responsibilities for preparedness functions, with particular attention to experiences in low- and middle-income settings [3,25]. It consolidates lessons from diverse contexts to provide an integrated overview of institutional strengths, weaknesses, and innovations, while highlighting underexplored dimensions such as autonomy, governance, and sustainable financing [21,25]. In doing so, the study bridges global frameworks with country-level realities and contributes evidence relevant to ongoing international initiatives, including the WHO Global Health Emergency Corps, aimed at strengthening resilient and surge-ready public health systems worldwide.

## Methods

### Study design

This scoping review was conducted following the methodological framework established by Arksey and O'Malley [12] and further refined by Levac et al [13] and was guided by the PRISMA-ScR checklist to ensure rigor and transparency. The review systematically mapped global and regional evidence on national-level Public Health Emergency Preparedness and Response (PHEPR), with emphasis on the role of National Public Health Agencies (NPHAs) in shaping governance,

workforce, and system-level capacities. The review sought to identify institutional arrangements, gaps, and transferable practices relevant to strengthening preparedness and response globally, with a particular focus on LMIC contexts. No protocol was registered.

### Research questions

This review was guided by the following questions:

1. How are National Public Health Agencies (NPHAs) and related institutions structured, and what mandates and roles do they hold in public health emergency preparedness and response (PHEPR)?

2. What governance arrangements, statutory authorities, and levels of autonomy enable or constrain NPHAs in leading and coordinating PHEPR?

3. How do NPHAs operationalize core PHEPR functions, including surveillance, laboratories, incident management, public health emergency operations centres (PHEOCs), risk communication, continuity of essential services, and countermeasure logistics?

4. What approaches are used to build and sustain skilled, surge-ready workforces for preparedness and response, and what challenges persist?

5. What financing models support preparedness, and how do countries address gaps in sustainable and predictable funding for PHEPR?

6. How are preparedness and response capacities extended and institutionalized at provincial, municipal, and community levels?

7. How do resilience, equity, digital interoperability, ethical safeguards, and community engagement influence the effectiveness and legitimacy of preparedness systems?

### Eligibility criteria

Studies were included if they met the following applied criteria:

• Publication period: Were published between January 2000 and April 2025.

• Scope: Focused on national-level or multi-country institutional arrangements for Public Health Emergency Preparedness and Response (PHEPR), or were explicitly aligned with the WHO Mapping of National Preparedness and Response Capabilities instrument.

• Type of evidence: Included peer-reviewed articles, validated survey submissions, institutional assessments, or technical reports providing structured and verifiable analysis.

• Thematic focus: Addressed one or more domains, including governance and autonomy, workforce development, financing and sustainability, surveillance and laboratory systems, risk communication and community engagement, or subnational preparedness structures.

• Language: Were published in English only.

 Studies were excluded if they:

• Focused exclusively on local, facility-specific, or project-based interventions without a clear linkage to national systems or policies;

---

- Lacked methodological transparency or sufficient analytical depth; or

- Consisted of commentaries, opinion pieces, or news reports without empirical grounding.

## Information sources and search strategy

Two primary evidence streams informed this review. The first consisted of peer-reviewed publications, while the second included grey literature retrieved from major databases and institutional sources. Searches were conducted across PubMed, Scopus, Web of Science, Cochrane Library, CINAHL, and selected national public health agency websites to capture diverse evidence on national-level public health emergency preparedness and response (PHEPR). Studies published from January 2000 through April 2025 were eligible for inclusion. The year 2000 was selected to align with the modern phase of public health emergency preparedness, characterized by IHR implementation and the expanding role of National Public Health Institutes as core national mechanisms for preparedness and response [20,26–28]. This timeframe reflects the modern era of institutionalized public health emergency preparedness and response.

Database searches employed both controlled vocabulary and free-text terms. In PubMed, Medical Subject Headings (MeSH) were combined with keywords such as *"National Public Health Institute," "public health emergency management," "emergency preparedness," "response governance," "EPR functions,"* and *"NPHA roles."* Equivalent terms were adapted for Web of Science using the Topic Search (TS) function and for Scopus using TITLE-ABS-KEY fields. Boolean operators (AND/OR) were used to link preparedness, workforce, and institutional terms. Searches were limited to English-language, human-related studies published between 2000 and 2025. Grey literature searches applied file-type filters (e.g., *filetype:pdf*), document-type keywords (*report, framework, guideline, policy*), and organizational site restrictions. Google Scholar was systematically searched using the same terms, and only the first 200 results were screened for feasibility. The full search strategies, filters, and limits are presented in S1 Table. The search strategy incorporated both terms, including "National Public Health Institute," "National Public Health Agency," and related descriptors. Studies referring to either institutional model were eligible, provided they addressed national entities with responsibility for preparedness and response functions.

## Selection process

All retrieved records were imported into EndNote for deduplication prior to screening. Screening was conducted in two sequential phases. First, titles and abstracts of all records were reviewed against predefined eligibility criteria. Second, articles considered potentially relevant were advanced to full-text assessment. Two reviewers (SDS and FMA) independently conducted both phases of screening using a structured and standardized process.

For each article assessed at full text, an explicit inclusion or exclusion decision was recorded, together with a specific rationale for exclusion where applicable. Discrepancies between reviewers were resolved through discussion and consensus. To ensure transparency and reproducibility, a comprehensive inventory of all full-text assessed studies was maintained, documenting inclusion status and reasons for exclusion. This inventory is presented in S2 Table.

To promote consistency in screening decisions, a calibration exercise was performed prior to full screening. A random sample of ten records was independently reviewed by both reviewers at the title and abstract stage. Agreement was reached on nine of ten records, corresponding to a raw agreement rate of 90% and a Cohen's kappa statistic of 0.82, indicating substantial inter-rater reliability [29]. Following calibration, the remaining records were screened independently by the two reviewers. Any disagreements were resolved through discussion, and unresolved cases were adjudicated by a third reviewer (ZMA). The overall screening process is summarized in the PRISMA-ScR flow diagram (Fig 1).

## Data extraction

Data extraction followed a standardized and piloted framework specifically developed for this review. Each included record was reviewed in detail, and information was charted across seven domains: authors and year of publication; continent or

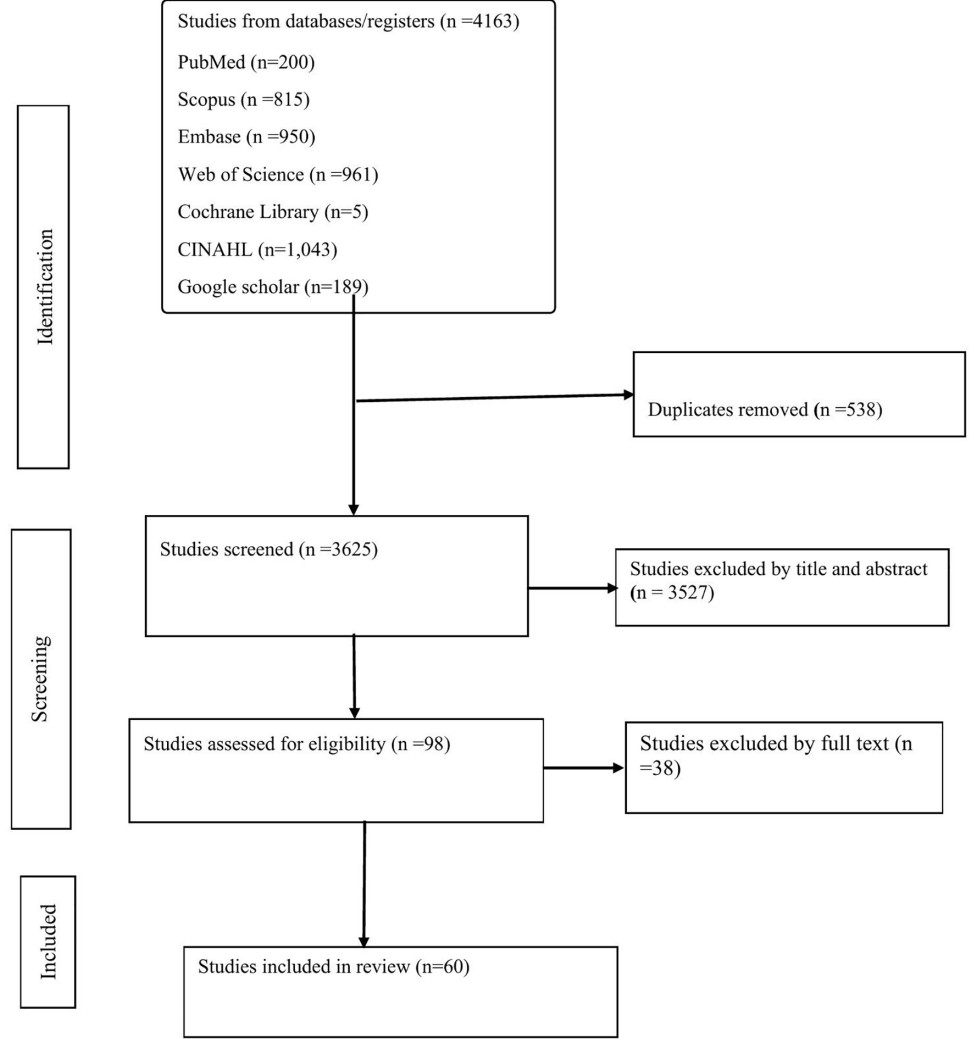

**Fig 1. PRISMA flow diagram of study selection.** This figure illustrates the identification, screening, eligibility assessment, and final inclusion of studies in the scoping review.

region; type of study or document; areas of focus; pertinent findings; identified gaps or study limitations; and recommendations. The framework was piloted on five records and refined to improve clarity, comprehensiveness, and analytical precision. In particular, the *"areas of focus"* domain was broadened to capture dimensions such as competencies, training, governance, and technology, while the distinction between *"recommendations"* and *"best practices"* was emphasized to enhance interpretive value.

Data extraction was conducted independently by two reviewers (SDS and FMA) using a standardized extraction framework developed for this review. Discrepancies in extracted information were resolved through discussion and consensus, and where agreement could not be reached, a third reviewer (SAS) provided adjudication. All relevant data from included studies were systematically recorded according to predefined fields to ensure consistency and completeness. The structured extraction dataset is provided in S4 Table. Consistent with scoping review methodology, formal risk-of-bias or quality appraisal of included studies was not performed. The purpose of this review was to map the scope and characteristics

of available evidence rather than to assess intervention effectiveness. Methodological limitations reported by the original authors were extracted and summarized to support transparent interpretation of findings.

## Data analysis and synthesis

An inductive thematic analysis approach was used to synthesize data extracted from the included studies. After data charting, all extracted information was systematically reviewed, coded, and organized into analytical categories through an iterative process of comparison and refinement. This approach allowed evidence from diverse study designs and document types to be examined in a consistent and structured manner. The analysis aimed to move beyond descriptive aggregation and to generate an interpretive synthesis of institutional arrangements, governance mechanisms, and operational practices relevant to national public health emergency preparedness and response (PHEPR). Analytical themes were developed inductively from the data and were subsequently used to organize the presentation of results.

## Critical appraisal

In line with the objectives of a scoping review, no formal risk-of-bias or methodological quality assessment was conducted. The primary purpose of this review was to map the scope, characteristics, and content of available evidence on national PHEPR systems rather than to evaluate the internal validity of individual studies. During data extraction, information on study design, data sources, and reported limitations was systematically recorded to support transparent interpretation. All eligible sources were included regardless of methodological rigor in order to capture the full range of perspectives and evidence relevant to the review objectives.

## Handling of missing data

The review relied exclusively on information reported in published and publicly available sources. Where predefined extraction variables were not described in an included study, these fields were recorded as "not specified." No attempts were made to contact authors or to impute missing information, in keeping with the descriptive and mapping objectives of a scoping review. Variations in reporting completeness across studies were considered during analysis, and findings were interpreted cautiously when data were limited or incomplete.

# Result

## Study selection

A total of 4,163 records were identified from databases/registers PubMed (n = 200), Scopus (n = 815), Embase (n = 950), Web of Science (n = 961), Cochrane Library (n = 5), CINAHL (n = 1,043), and Google Scholar (n = 189). After removal of 538 duplicates, 3,625 records remained for title/abstract screening, of which 3,527 were excluded. 98 full-text articles were assessed for eligibility; 38 were excluded (irrelevant to objectives, insufficient information for extraction, or outside scope). Sixty studies were included in the final synthesis. The selection process is depicted in the PRISMA flow diagram (Fig 1).

## Study characteristics

A total of 60 studies met the inclusion criteria following full-text screening. The search strategy covered the period from 2000 to 2025; however, the earliest eligible publication identified was from 2007. The included studies therefore span the years 2007–2025 and represent global, regional, and national contexts. Early publications (2007–2015) were primarily conceptual, focusing on defining public health emergency preparedness (PHEP) frameworks and governance models. Research output expanded during 2016–2019 with the introduction of resilience frameworks, preparedness assessment tools, and capability standards. A marked increase occurred during the COVID-19 period (2020–2022), reflecting intensified attention to digital preparedness, genomic surveillance, public health and social measures (PHSMs), and system

resilience. In the post-COVID period (2023–2025), the literature shifted toward governance arrangements, financing mechanisms, institutional autonomy, and empirical analyses from low- and middle-income countries (LMICs), particularly in Africa.

Geographically, 32 studies had a global scope, while others originated from Europe (7), Africa (7), Asia (6), the Americas (5), and the Eastern Mediterranean Region (3). Five studies were conducted in Ethiopia, providing detailed empirical evidence on surveillance performance, inter-institutional coordination, and preparedness assessment. Methodologically, the review included empirical or program evaluations (n = 17), framework or tool-development papers (n = 15), reviews or scoping syntheses (n = 13), and commentaries or policy analyses (n = 15). Thematically, studies clustered around preparedness and response functions, institutional governance and autonomy, workforce and surge capacity, financing and sustainability, and cross-sectoral coordination, forming the evidence base for the seven analytical themes synthesized in this review. Detailed characteristics, key findings, limitations, and recommendations of all included studies are provided in S4 Table.

### Identified themes from the scoping review

The evidence synthesized in this review was organized into seven interrelated themes that define the institutional and functional landscape of public health emergency preparedness and response (PHEPR). These themes are: (1) Institutional Landscape, (2) Governance and Autonomy, (3) Preparedness and Response Functions, (4) Workforce Development, (5) Financing, (6) Subnational Presence, and (7) Cross-Cutting Considerations.

**Theme 1: Institutional landscape.** The reviewed studies document a global expansion of institutional arrangements for public health emergency preparedness and response (PHEPR), particularly through the establishment and strengthening of national public health institutes (NPHIs). Empirical analyses report that, following the COVID-19 pandemic, NPHIs in many countries assumed expanded responsibilities in national surveillance, emergency coordination, and outbreak response activities [30]. Earlier literature similarly describes the creation of NPHIs as a structural strategy to consolidate public health functions and to support national implementation of the International Health Regulations (2005) [31]. A global scoping review found that NPHIs are now present in numerous countries, but with substantial variation in mandate, autonomy, and operational maturity [32].

Regional studies provide additional evidence of diverse institutional models. Research from the Eastern Mediterranean Region reports the establishment of NPHIs in several countries, while also noting limitations in statutory authority and domestic financing [33]. Studies from fragile health systems describe preparedness responsibilities as frequently distributed across multiple ministries and agencies, resulting in varied coordination arrangements [3]. An assessment from Ghana identified defined national preparedness institutions but reported challenges in cross-sectoral integration [34]. A qualitative study from Palestine similarly documented divided institutional authority for emergency response among several governmental bodies [35]. In contrast, literature from the United States describes preparedness capability standards supported by established legal frameworks and long-term institutional investment [36].

European analyses describe preparedness systems involving multiple sectors, with NPHIs functioning as technical coordinating bodies within broader governmental arrangements [37]. Historical accounts from Brazil report the integration of emergency management functions into national health reforms, creating organizational linkages between preparedness structures and routine health institutions [38]. Additional studies note that established systems have undergone organizational adjustments in response to evolving threats [39,40].

Across the literature, empirical findings consistently highlight differences between contexts. Studies from high-income countries describe structured preparedness frameworks with stable institutional arrangements [30,36,40]., while studies from low- and middle-income countries more frequently document fragmented authority structures and reliance on external funding [3,33–35]. Country-level assessments from Ethiopia, Ghana, and Palestine further illustrate variability in governance, coordination, and financing for PHEPR [34,35,41–44].

**Theme 2: Governance and autonomy.** The reviewed studies describe diverse governance arrangements that structure public health emergency preparedness and response (PHEPR) systems. Empirical analyses document statutory mandates, formal reporting lines, and defined decision-making authorities within national preparedness frameworks [9,10,24,36]. Several countries are reported to have codified roles for incident command systems and emergency operations centers as part of routine governance arrangements [9,10,24,36]. Additional studies describe the incorporation of the International Health Regulations (2005) into domestic legal and institutional mechanisms [45]. Where governance relies on ad hoc directives or fragmented structures, studies report variation in activation procedures, risk communication, and resource allocation across agencies [3,34,35]. Post-COVID analyses describe governance systems that combine political decision-making mechanisms with formal processes for the use of scientific evidence in emergency actions [46–48].

Institutional autonomy is consistently identified as a distinguishing feature of preparedness systems. Studies describe differences between agencies with formally protected technical authority and those operating under closer political control [30,31,36,40]. Comparative reviews report governance frameworks that include statutory mandates, delegated procurement powers, and formal coordination authority across sectors [37,45,48,49]. Country-level assessments from Ghana and Palestine document overlapping institutional mandates and constrained operating space for national preparedness bodies [34–36]. Jurisdictions with longer-established standards are described as demonstrating closer alignment between governance structures, accountability mechanisms, and financing arrangements [36].

The literature further documents the use of specific governance tools. The "Ready, Willing, and Able" framework is described as an approach for assessing organizational readiness [11]. Analyses from the United States report formal capability standards specifying functions for incident management and coordination [7]. Accreditation programs are reported to support structured documentation and quality improvement processes related to preparedness [50,51]. Regional studies describe logic models used to define accountability arrangements for cross-border threats [10]. Global mechanisms such as GOARN and surveillance platforms including EWARS and EWARN are described as operating through formal governance agreements covering data stewardship and activation protocols [52–54].

Additional studies address governance processes related to ethics and service continuity. Reviews describe institutional provisions for transparency, privacy protection, and formal procedures for evidence use in decision-making [39,46,47,55]. Analyses of essential service continuity report that clearer governance roles are associated with adaptive service delivery models [56,57]. Post-event studies document reforms introduced after emergencies, including mandate clarification and formalization of multi-sectoral coordination mechanisms [38,48,58–61]. Empirical studies from Ethiopia, Ghana, and Palestine further describe overlapping mandates and variability in fiscal and operational autonomy across national and subnational levels [41,42,62].

**Theme 3: Preparedness and response functions.** The reviewed studies describe national preparedness and response systems as structured around a defined set of operational functions, including incident management, public health emergency operations centers (PHEOCs), surveillance and analytics, laboratory diagnostics, risk communication and community engagement, public health and social measures (PHSM), continuity of essential services, and logistics management. Capability frameworks and program evaluations document efforts to formalize these functions through standardized protocols, defined roles, and structured reporting systems [1,2,5,22,36]. National and regional assessments report the use of incident command structures, situation reports, and after-action review processes as routine components of preparedness systems [5,36]. Several studies describe mechanisms to support evidence-informed decision-making, including rapid reviews, decision logs, and structured evaluation processes [47,63]. Other analyses document the integration of simulation exercises and corrective-action tracking within preparedness programs [48,52].

Surveillance and analytic capacity are consistently identified as core elements of preparedness. Studies describe the deployment of interoperable platforms such as EWARS and EWARN to support early warning and event-based surveillance, dependent on agreed data standards and stewardship arrangements [48,52]. Global mechanisms, including the

Global Outbreak Alert and Response Network (GOARN), are reported to operate through predefined activation protocols and coordinated deployment systems [48,49]. Additional accounts describe the expansion of analytic practices to include modelling and forecasting units embedded within emergency operations structures [4, 64]. Evaluations of genomic surveillance during the COVID-19 pandemic document increased sequencing capacity alongside variability in turnaround times and integration with operational dashboards [65,66].

Risk communication and community engagement are described as formal operational components within emergency systems. Competency frameworks outline structured approaches to social listening and message development, and several studies report the establishment of dedicated RCCE units within PHEOCs [12,67]. Logistics and countermeasure management are similarly presented as foundational functions, supported by inventory-visibility tools, diversified procurement channels, and standardized operational protocols [46,48,68]. Subnational assessments describe the use of local preparedness scorecards to operationalize national capability standards [13,69].

**Theme 4: Workforce and capacity development.** The reviewed studies consistently document the central role of workforce capacity in public health emergency preparedness and response (PHEPR). Empirical analyses identify Field Epidemiology Training Programs (FETPs) and structured workforce assessment tools as commonly used mechanisms for developing national preparedness capacity [6,11,51,63,70]. Studies report that these programs provide training in surveillance, analytic methods, and incident management functions, and are incorporated into professional development pathways in multiple countries. In high-income settings, preparedness frameworks describe workforce metrics that include roster completeness, surge readiness, and deployability as measurable components of national capability standards [36,51]. In contrast, studies from low- and middle-income countries document challenges related to fragmented human resource planning, limited funding for training, and dependence on externally supported initiatives [3,33,34].

Surge deployment capacity is described as a key element of workforce systems. Empirical reports document the use of pre-arranged surge mechanisms such as rapid response teams, global rosters, and formally trained emergency cadres in several countries [6,7]. Evaluations of national and international response teams describe modular training programs, defined deployment protocols, and interoperability arrangements across sectors [7,53]. Studies from Ghana and Kenya report that surge deployment efforts were affected by overlapping institutional mandates, resource constraints, and limitations in financing arrangements [12,34].

Multiple studies also report on workforce sustainability. Empirical analyses describe challenges related to staff attrition, burnout, and limited career progression, particularly in contexts where preparedness positions are funded through short-term contracts or external sources [3,33,35]. Community-level studies document that frontline worker in several countries experienced limited access to psychosocial support, formal recognition, and occupational protection during prolonged emergency responses [12]. Framework analyses describe approaches for integrating workforce planning within national human resource strategies, including the use of monitoring tools and accreditation-linked incentives [51,63,70]. Country reports from Brazil and the United States describe arrangements that combine technical training programs, legal protections for responders, and structured career development pathways [36,38].

Country-level empirical studies provide additional descriptive evidence on workforce capacity. Research from Ethiopia, Ghana, and Kenya documents variability in workforce preparedness across institutional levels and administrative tiers [12,34,41–44]. Facility-based assessments report differences in staff readiness, availability of trained personnel, and implementation of core preparedness competencies [43]. National evaluations describe gaps in specimen transport, reporting timeliness, and use of surveillance data within emergency operations centers [44]. Other studies report challenges related to information sharing, digital interoperability, and coordination among institutions involved in emergency response [62].

**Theme 5: Financing.** The reviewed studies document financing as a central component of public health emergency preparedness and response (PHEPR) systems. Empirical analyses report that many countries rely on external or short-term funding sources to support preparedness activities [3,33,34,58,59]. Several studies describe patterns in which donor

resources increase during emergencies and decline after acute phases, affecting sustained investment in laboratories, surveillance systems, and workforce development [12,34,46]. Comparative research reports that in the absence of dedicated domestic budget lines, preparedness activities are frequently financed through ad hoc allocations or project-based mechanisms [31,45,48].

Multiple studies describe different financing approaches used to support preparedness. Empirical reports document the establishment of contingency funds, ring-fenced allocations, and performance-based financing mechanisms in some countries [58,59]. Case studies describe how these arrangements have been used to enable rapid disbursement of resources during emergencies and to support ongoing system functions. Other analyses report that many national systems continue to depend primarily on donor-supported or program-specific financing rather than institutionalized domestic funding streams [33,34]. Studies from low- and middle-income settings describe resource limitations that affect coordination, logistics, and surge response capacity [3]. Research from high-income contexts more frequently reports administrative and procedural challenges related to the management of preparedness funds [30,36,48].

Across the literature, empirical findings consistently describe differing financing patterns between countries and regions. Studies report that predictable, domestically sourced financing mechanisms are less common in lower-resource settings, while externally supported funding remains a major source of preparedness investment [3,33,34,45,48,58,59].

Country-level empirical studies provide additional descriptive evidence on financing arrangements. Research from Ethiopia, Ghana, and other settings documents variability in budget predictability and financial autonomy among institutions responsible for emergency management [34,42–44,71]. Assessments from Ethiopia report differences in the availability of contingency budgets across regional and local administrative levels [71]. Framework analyses describe reliance on externally funded programs to support preparedness activities and system strengthening initiatives [42]. Facility-level evaluations report that delayed fund release and limited operational budgets affected training, logistics, and surveillance activities [43]. National surveillance evaluations document resource constraints influencing specimen transport, testing capacity, and routine preparedness functions [44]. These findings are consistent with reports from other low- and middle-income countries describing dependence on external funding and variability in domestic financing for PHEPR [3,33,34].

**Theme 6: Subnational presence.** The reviewed studies document that public health emergency preparedness and response (PHEPR) systems operate across national, regional, and local administrative levels. Empirical analyses describe regional and local health departments as primary sites for event detection and initial response activities. Studies from Ghana and Kenya report that while national policies establish preparedness frameworks, subnational units often operate with limited resources, fragmented authority structures, and constrained surge capacity [12,34]. Research from Palestine similarly documents institutional fragmentation at local levels and describes coordination processes that vary across districts and agencies [35].

Municipal and urban preparedness studies provide additional descriptive evidence on subnational arrangements. Assessments from several countries report that densely populated urban settings require coordination among multiple local institutions, including municipal authorities, health departments, and emergency services [13,69]. Evaluations describe municipalities that maintain local emergency plans, stockpiles, and integrated warning systems as having structured operational arrangements [13,69]. Other studies document that rural and peri-urban areas frequently have more limited logistics infrastructure and health service access compared with major urban centers [3,12].

Across the literature, empirical findings describe differing relationships between national and subnational preparedness structures. Studies report that local authorities in some settings operate with defined operational responsibilities and budgetary discretion, while in other contexts decision-making remains more centralized [36,37,45]. Additional analyses document that the implementation of national preparedness directives at provincial and district levels varies according to available resources, staffing, and coordination mechanisms [3,13,34–37,45,69].

Country-level empirical studies provide comparable evidence on subnational preparedness. Research from Ethiopia, Ghana, and Kenya documents variability in preparedness capacity across regional and district levels [12,34,41–44]. In Ethiopia, validation of the PHEM Assessment Tool identified differences among regions in coordination structures, logistics capacity, and financing arrangements [41]. Framework analyses from the same context describe limited operational autonomy for regional and zonal structures and reliance on central decision-making processes [42]. Facility-based studies from Addis Ababa report that sub-city health offices and health centers experienced constraints related to emergency funding, surveillance feedback, and reporting channels [43]. National surveillance evaluations further document inconsistencies in coordination between regional laboratories and national reference centers, affecting specimen transport and response processes [44].

**Theme 7: Cross-cutting considerations.** The reviewed studies describe several cross-cutting factors that influence public health emergency preparedness and response (PHEPR) systems. Empirical analyses report that preparedness frameworks in multiple countries incorporate concepts of adaptability, redundancy, and structured feedback mechanisms within institutional design [48,58,59,61]. Studies examining public health and social measures (PHSM) document the inclusion of community engagement activities, risk communication processes, and mechanisms intended to support continuity of essential services [46,48,56,68]. Additional research describes the impact of preparedness actions on different population groups and reports that service disruptions during emergencies have varied across communities and social contexts [56,57].

Digitalization and interoperability are frequently described in the empirical literature as important components of preparedness systems. Multiple studies document the expansion of digital surveillance platforms and genomic sequencing capacities during recent emergencies [64–66]. Evaluations report differences between countries in access to digital tools, data-sharing arrangements, and the integration of information systems across agencies [52,64,65]. Additional analyses describe the use of digital and social listening platforms for risk communication and infodemic management, with varying levels of institutionalization across settings [12,67].

Partnerships and ethical processes are also reported as cross-cutting elements of preparedness. Studies describe global mechanisms such as the Global Outbreak Alert and Response Network (GOARN) as operating through predefined agreements and interoperable systems for coordination [7,54]. Empirical reviews document governance arrangements that include provisions for transparency, proportionality, and formal use of evidence in emergency decision-making [39,46,47,55]. Across the literature, preparedness systems are described as involving interactions between institutional structures, operational processes, and community-level engagement mechanisms [37,46–48,52,55,56,59,61,65–68].

Country-level empirical studies provide additional descriptive evidence on these cross-cutting factors. Research from Ethiopia, Ghana, and other settings reports variability in digital interoperability, information governance, and communication systems [39–41,60]. Studies from Ethiopia describe limitations in data exchange among national institutions involved in emergency response and document differences in information-management capacity across administrative levels [41,42,62]. Facility-based assessments report inconsistencies in communication channels and community engagement practices at local levels [43]. Similar findings from other low- and middle-income countries describe gaps in integration between surveillance, laboratory, and logistics information systems, as well as differences in the availability of trained information-technology personnel [12,64,67,72].

## Thematic synthesis of findings

The seven-theme synthesis (Table 1) summarizes the empirical findings reported across the reviewed studies on public health emergency preparedness and response (PHEPR) systems. The literature describes institutional arrangements centered on National Public Health Institutes (NPHIs) that operate within varying legal, administrative, and financing structures. Studies document that preparedness functions are organized through codified operational procedures, technical platforms, and defined coordination mechanisms, with implementation occurring at both national and subnational levels.

**Table 1. Identified themes from the scoping review on public health emergency preparedness and response.**

| Theme | Key Empirical Insights | Policy and System Implications |
|---|---|---|
| 1. Institutional Landscape | National Public Health Institutes (NPHIs) have expanded globally as central preparedness institutions. However, maturity, mandates, and autonomy vary widely. Performance depends on legal entrenchment, political legitimacy, and integration with multisectoral systems. | Preparedness strategies should prioritize institutional consolidation rather than creating parallel structures. Legal mandates and sustainable core financing are more critical than expanding new technical units. |
| 2. Governance & Autonomy | Systems with clear statutory authority, defined decision rights, and protected technical independence act more quickly and credibly. Fragmented mandates and politicized oversight weaken coordination and erode continuity. | Strengthening governance frameworks is foundational. Clarifying roles, codifying emergency authorities, and protecting scientific independence are essential to operational readiness. |
| 3. Preparedness & Response Functions | Core capacities—PHEOCs, surveillance, laboratories, logistics, RCCE, PHSMs, and service continuity—perform best when codified, interoperable, and supported by routine exercises and evidence-based decision processes. | Investments should focus on integration and standardization rather than creating new vertical programs. Regular simulations, after-action reviews, and interoperable platforms are critical for functionality. |
| 4. Workforce & Capacity Development | Workforce readiness depends on structured training pathways (e.g., FETPs), surge rosters, and retention systems. In many LMICs, capacity remains donor-driven, fragmented, and vulnerable to attrition. | Preparedness requires long-term human resource strategies, protected staffing lines, and institutionalized career pathways rather than episodic trainings or project-based hiring. |
| 5. Financing | Reliance on external or crisis-triggered funding creates cycles of surge and neglect. Few countries have operational contingency funds or predictable preparedness budgets. | Sustainable domestic financing is a central enabler of resilience. Ring-fenced preparedness budgets and rapid disbursement mechanisms should replace ad hoc donor dependence. |
| 6. Subnational Presence | Detection and response are inherently local, yet regional and municipal structures often lack authority, resources, and coordination mechanisms. Excessive centralization limits responsiveness. | Effective preparedness requires empowered subnational systems with delegated authority, operational budgets, and integrated reporting channels to national structures. |
| 7. Cross-Cutting Considerations | Digital interoperability, community engagement, ethical governance, and equity shape outcomes across all domains. Fragmented information systems and weak trust undermine technical capacities. | Preparedness reforms must embed interoperability standards, transparent communication, and equity safeguards to ensure legitimacy, inclusiveness, and public confidence. |

Evidence from multiple countries reports differences in workforce capacity, surge deployment systems, and the availability of sustainable financing. Additional findings describe variability in local implementation of national directives, as well as differing levels of autonomy and resource availability across administrative tiers. Cross-cutting dimensions, including digital interoperability, information governance, ethics, and community engagement, are consistently reported as components of preparedness systems. Collectively, these empirical results provide a structured overview of how PHEPR capacities are organized, implemented, and experienced across diverse global contexts, forming the basis for the subsequent Discussion (Table 1).

## Discussion

This scoping review set out to understand how public health emergency preparedness and response (PHEPR) systems function in practice. Using a structured synthesis of 60 studies, the review organized global evidence into seven domains: institutional arrangements, governance, preparedness functions, workforce, financing, subnational implementation, and cross-cutting enablers [22,23,32,73,74]. Examining these domains together provides a systems-level view of

preparedness one that moves beyond individual tools or programs to consider how they operate as a connected whole. The collective findings indicate that preparedness effectiveness is shaped less by the presence of technical components than by the institutional conditions that enable those components to work in concert [14,21,31,32,45].

Across the literature, governance structures and institutional autonomy emerged as the central mechanisms that translate capacity into action. Studies repeatedly showed that systems function most effectively when public health institutions have clear legal mandates, defined decision rights, and operational independence [14,21,32,35,37]. Where authority was fragmented or roles were ambiguous, coordination faltered even when technical platforms were in place [33,35,37]. This pattern highlights a critical point: preparedness failures are often administrative rather than technical in nature. The effectiveness of surveillance systems, laboratories, and emergency operations centers depends fundamentally on whether institutions are empowered to lead, coordinate, and act [21,32,37]. Governance coherence, therefore, is not simply an organizational preference it is the backbone of functional preparedness [14,21,32,34,35,37].

The role of National Public Health Institutes provides an important lens for understanding these governance dynamics. The literature consistently positions NPHIs as the institutional anchors of preparedness, but the review indicates that their effectiveness depends less on organizational form than on statutory legitimacy, operational autonomy, and sustainable financing [5,30,33,47,69]. Institutions with protected technical authority and clear legal mandates are more able to coordinate sectors, translate evidence into decisions, and maintain continuity across political cycles. This reinforces the central conclusion that governance quality, rather than technical infrastructure alone, differentiates resilient systems from reactive ones.

Financing and workforce systems give practical expression to these governance arrangements. The evidence demonstrates that preparedness programs built on short-term or donor-driven funding are inherently fragile [23,24,34,40,45,57,59]. Cycles of emergency investment followed by neglect repeatedly erode surveillance capacity, laboratory networks, and response infrastructure [34,45,59]. Similarly, workforce findings show that training initiatives alone do not create readiness unless they are embedded within stable employment structures that preserve skills and institutional memory [6,11,68,70,73]. Preparedness, as described across the reviewed studies, functions best when it is treated as a routine public-sector responsibility supported by predictable budgets and professionalized human resources rather than as a temporary project activated only during crises [34,40,45]. This pattern has been described as a "surge-and-neglect" cycle in which emergency investments expand capacity temporarily but erode between crises when predictable domestic financing is absent [37,41,74].

The subnational dimension provides the clearest illustration of how system design shapes real-world outcomes. Emergencies are detected, investigated, and managed locally, yet many studies documented persistent gaps in authority, financing, and coordination at regional and district levels [12,13,34,35,37,57,69]. This mismatch between national planning and local implementation explains why comprehensive national frameworks often yield uneven performance on the ground. Preparedness is inherently a distributed function. National structures can set direction and standards, but operational success depends on whether provincial and municipal actors are equipped with the resources, information, and decision space needed to act [12,43,69]. Strengthening these local interfaces is therefore essential to closing the persistent gap between policy and practice.

Cross-cutting enablers particularly digital interoperability, community engagement, and ethical governance shape how the technical and institutional dimensions of preparedness come together. Digital tools have expanded rapidly, yet fragmented platforms and weak data governance frequently limit their operational value [41,49,52,54,56,58,59,64,66,67]. At the same time, studies consistently showed that public trust, transparent communication, and inclusive engagement are not optional add-ons but core determinants of compliance and effectiveness [10,36,50,67]. These factors remind us that preparedness is simultaneously technical, organizational, and social. Systems that integrate data, institutions, and communities are better positioned to convert plans into coordinated action [10,50,52,56,65,66,75]. Digitalization illustrates this tension particularly clearly. While surveillance and data platforms have expanded rapidly, fragmented systems and weak

interoperability frequently limit their operational value [14,44,45,48,53,59,60]. Equity and ethical governance play a similar enabling role: preparedness strategies that neglect trust, inclusion, and transparent communication consistently face implementation barriers even when technical capacity exists [7,12,36,60]. These interpretations align closely with contemporary global frameworks. Initiatives such as the WHO Global Health Emergency Corps, the Africa CDC NPHI Development Framework, and the Global Outbreak Alert and Response Network all emphasize empowered national institutions, interoperable systems, and sustainable financing as prerequisites for effective preparedness [4,11,50,76]. The convergence between these frameworks and the findings of this review underscores that strengthening institutional foundations is now widely recognized as the central pathway to resilient preparedness.

Taken as a whole, this review contributes to the field by reframing public health emergency preparedness as fundamentally institutional systems challenge rather than a primarily technical one. The synthesized evidence indicates that strategies focused narrowly on equipment, digital platforms, or short-term trainings rarely produce durable improvement unless they are embedded within coherent governance arrangements, predictable financing, professionalized workforces, and empowered subnational implementation [23,30–32,34,35,37,39,40,45]. By examining these domains together, the review demonstrates that preparedness performance emerges from the interaction of legal authority, organizational autonomy, resource stability, and operational linkages between national and local levels [14,21,32,34,35,37,39]. This integrated perspective helps explain why similar technical models generate different outcomes across contexts and clarifies that functional readiness depends less on the presence of capacities than on the institutional conditions that enable their effective use [33–35,37,39]. The principal contribution of this study is therefore to offer a systems-level synthesis that shifts attention from counting tools to strengthening the structural foundations required for continuously functional, equitable, and resilient preparedness.

## Strengths and limitations

This review has several strengths. It provides a comprehensive global synthesis of evidence on public health emergency preparedness and response using a transparent and systematic scoping review methodology. The inclusion of diverse study designs and geographical contexts enabled a broad systems-level analysis that integrates institutional, operational, and policy dimensions. The structured seven-domain framework offers a coherent approach to examining how governance, financing, workforce capacity, subnational implementation, and cross-cutting factors interact to shape preparedness performance.

The study also has limitations. As a scoping review, it maps the breadth of available evidence but does not formally assess methodological quality or evaluate the effectiveness of specific interventions. The findings rely on accessible published sources, which may underrepresent unpublished program experiences and rapidly evolving practices. Inclusion of grey literature strengthened the review by capturing national assessments, policy documents, and operational reports that are essential to understanding real-world preparedness, particularly in low- and middle-income settings. However, grey literature varies in methodological transparency and analytical rigor, introducing heterogeneity into the evidence base and requiring cautious interpretation. Differences in terminology and reporting across studies further limited direct comparability, and contextual variations between countries restrict the generalizability of some observations.

## Policy implications

The findings of this review indicate that strengthening public health emergency preparedness requires a clear shift in policy priorities.

First, preparedness should be grounded in clear governance arrangements. Immediate gains can be achieved by clarifying legal mandates, defining decision rights, and formalizing interagency coordination mechanisms. Technical capacity delivers results only when institutions are authorized and accountable to lead.

Second, financing for preparedness must become routine and predictable. Establishing dedicated budget lines, contingency funding mechanisms, and pre-approved emergency procedures offers a practical pathway to sustain surveillance, laboratories, and emergency operations between crises.

Third, workforce readiness requires long-term institutionalization. Countries can strengthen preparedness by formalizing surge rosters, integrating Field Epidemiology Training Program graduates into permanent positions, and embedding continuous professional development within national human resource systems.

Fourth, subnational systems should be deliberately empowered. Regional and municipal levels need defined authority, modest operational budgets, and structured reporting channels so that national strategies translate into effective local action.

Fifth, digital investments should prioritize interoperability over expansion. Harmonized data standards, shared platforms, and clear information-governance arrangements are essential to link surveillance, laboratories, and emergency operations into a single operational environment.

Sixth, preparedness benefits from routine learning processes. Regular simulation exercises, after-action reviews, and corrective action tracking should be institutionalized to maintain readiness and improve performance over time.

Finally, international assistance is most effective when it reinforces national systems. Support that strengthens domestic institutions and subnational capacity provides more durable impact than short-term, parallel surge responses.

Together, these priorities point to a central message: preparedness improves not by adding new tools, but by enabling institutions to use existing tools effectively through clear authority, stable resources, professionalized workforces, and empowered local implementation.

## Conclusions

This scoping review mapped global evidence on public health emergency preparedness and response and identified the institutional and operational conditions that shape system performance. Across the reviewed literature, preparedness was most effective where public health institutions functioned with clear legal mandates, predictable financing, supported workforces, and empowered subnational structures. Persistent weaknesses were consistently associated with fragmented governance, donor-dependent funding, limited local implementation, and inadequate digital interoperability. By synthesizing findings across seven interrelated domains, the review demonstrates that the primary determinants of preparedness are institutional and systemic rather than purely technical.

In summary, preparedness improves not by adding more tools, but by enabling institutions to act. Policies that strengthen governance clarity, sustainable financing, workforce stability, and subnational implementation provide the most reliable pathway to resilient and continuously functional preparedness systems.

## Declarations

### Patient and public involvement

Patients and members of the public were not directly involved in the design, conduct, reporting, or dissemination of this scoping review. The study synthesized existing published evidence on public health emergency preparedness and response. Nevertheless, the findings are intended to inform policy and practice in ways that ultimately strengthen health systems and benefit communities.

## Supporting information

**S1 Checklist. PRISMA-ScR checklist.** The PRISMA-ScR checklist is reproduced from Tricco et al. (2018) under the Creative Commons Attribution 4.0 International License (CC BY 4.0).
(DOCX)

**S1 Table. Search strategy across databases.** This table presents the full electronic search strategies used across PubMed, Scopus, Web of Science, EMBASE, CINAHL, Cochrane Library, and Google Scholar.
(DOCX)

**S2 Table. Inventory of full-text assessed studies.** This table lists all studies assessed at full-text stage (n = 98), indicating included and excluded articles with specific reasons for exclusion.
(DOCX)

**S3 Table. Detailed data extraction for included studies.** This table provides structured data extraction for the 60 included studies, including study characteristics, key findings, limitations, policy implications, and analytical domains.
(DOCX)

**S4 Table. Summary characteristics and findings of included studies.** This table synthesizes characteristics, pertinent findings, identified gaps, and recommendations across all included studies.
(DOCX)

## Acknowledgments

The authors gratefully acknowledge the Ethiopian Public Health Institute (EPHI) for facilitating this research. We also thank colleagues and partner institutions who provided background insights, access to documents, and technical guidance during the review and synthesis phases.

## Author contributions

**Conceptualization:** Sileshi Demelash Sasie, Tigist Ali Gebeyehu, Zenebech Mamo Argaw.

**Data curation:** Fantu Mamo Aragaw, Zenebech Mamo Argaw.

**Formal analysis:** Sileshi Demelash Sasie, Fantu Mamo Aragaw, Tigist Ali Gebeyehu, Sintayehu Abdella Shikur, Lensa Fekadu.

**Investigation:** Sileshi Demelash Sasie.

**Methodology:** Sileshi Demelash Sasie, Fantu Mamo Aragaw, Sintayehu Abdella Shikur, Neima Zeynu Ali, Zenebech Mamo Argaw.

**Resources:** Sileshi Demelash Sasie.

**Software:** Tigist Ali Gebeyehu, Neima Zeynu Ali, Zenebech Mamo Argaw.

**Supervision:** Lensa Fekadu, Neima Zeynu Ali.

**Validation:** Fantu Mamo Aragaw, Sintayehu Abdella Shikur, Zenebech Mamo Argaw.

**Visualization:** Sintayehu Abdella Shikur, Lensa Fekadu, Zenebech Mamo Argaw.

**Writing – original draft:** Sileshi Demelash Sasie, Tigist Ali Gebeyehu, Sintayehu Abdella Shikur, Lensa Fekadu, Zenebech Mamo Argaw.

**Writing – review & editing:** Sileshi Demelash Sasie, Tigist Ali Gebeyehu, Lensa Fekadu, Neima Zeynu Ali, Zenebech Mamo Argaw.

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
