## [Decision Letter · Decision Letter 0]

6 Jan 2026

PGPH-D-25-03097

Building Systems for Preparedness: Global Scoping Studies on Institutional Governance and National Public Health Agencies

Dear Dr. Sasie,

Thank you for submitting your manuscript to PLOS Global Public Health. After careful consideration, we feel that it has merit but does not fully meet PLOS Global Public Health’s publication criteria as it currently stands. Therefore, we invite you to submit a revised version of the manuscript that addresses the points raised during the review process.

The manuscript has been evaluated by three reviewers, and their comments are available below.

The reviewers have raised a number of concerns that need attention. They request additional information on methodological aspects of the study,  clarification on the difference between NPHA and NPHI (if any in the extent of the paper, and state in the results whether these are applicable to one or both. The reviewers also request for the discussion to be re-balanced (currently much focus is on Ethiopia without justification why this is so)

Could you please revise the manuscript to carefully address the concerns raised?

We look forward to receiving your revised manuscript.

Kind regards,

Katrien G. Janin, PhD

Staff Editor

Journal Requirements:

1. Please confirm the parameters for the literature included in your scoping review. We note that the abstract states literature between 2007 and 2025 were included. In the methods, the manuscript mentioned including literature from 2000 to 2025 in the “Eligibility” section and literature from 2015 to 2025 in the “Information Sources and Search Strategy” section.

2. As required by our policy on Data Availability, please ensure your manuscript or supplementary information includes the following:

3. Please amend your online Financial Disclosure statement. If you did not receive any funding for this study, please simply state: “The authors received no specific funding for this work.”

4. Please update your online Competing Interests statement. If you have no competing interests to declare, please state: “The authors have declared that no competing interests exist.”

5. Please provide separate main figure files in .tif or .eps format only and ensure that all files are under our size limit of 10MB.

6. Please include a separate legend or caption for Figure 1 in your manuscript.

7. Main tables should not be uploaded as individual files. Please remove these files and include the Tables in your manuscript file as editable, cell-based objects. For more information about how to format tables, see our guidelines: https://journals.plos.org/globalpublichealth/s/tables

Please note that supplementary tables should remain uploaded as separate 'Supporting Information' files.

8. “Supplementary materials.docx” is currently uploaded as an ‘Other’ file type, which is not viewable by reviewers. Please ensure that all files meant for review are uploaded as 'Supporting Information' and include a legend in the manuscript.

Additional Editor Comments (if provided):

Reviewers' comments:

Reviewer's Responses to Questions

**Comments to the Author**

1. Does this manuscript meet PLOS Global Public Health’s publication criteria?

Reviewer #1: Yes

Reviewer #2: Yes

2. Has the statistical analysis been performed appropriately and rigorously?

Reviewer #1: Yes

Reviewer #2: Yes

3. Have the authors made all data underlying the findings in their manuscript fully available (please refer to the Data Availability Statement at the start of the manuscript PDF file)?

Reviewer #1: Yes

Reviewer #2: Yes

4. Is the manuscript presented in an intelligible fashion and written in standard English?

Reviewer #1: Yes

Reviewer #2: Yes

Reviewer #1: Major Points for Revision

1. Length and redundancy

The manuscript is comprehensive but would benefit from tightening. There is substantial overlap between the Results and Discussion sections, with repeated restatement of findings and citations.

Recommendation:

o Streamline the Results section to focus on empirical patterns (What was found”).

o Refocus the Discussion on interpretation, implications, and contribution (“why it matters”).

o Reducing redundancy would substantially improve readability and editorial fit.

2. Sharpen the central analytical thesis

While the core insight is clear, it is diffused across sections.

Recommendation:

o Explicitly articulate the central argument early in the Introduction and reiterate it at the start of the Discussion, for example: preparedness effectiveness depends less on the existence of institutions than on their integration within legally mandated, autonomous, and sustainably financed governance systems with empowered subnational implementation.

3. Methods section consistency

There are minor inconsistencies in the reported publication date ranges across sections (i.e., 2007-2025 in Abstract, 2000-2025 in Methods).

Recommendation:

o Harmonize and clearly justify a single inclusion period to avoid confusion for readers and reviewers.

4. Terminology consistency

Multiple related terms (PHEP, PHEPR, EPR) are used interchangeably.

Recommendation:

o Standardize terminology (e.g., PH EPR) and define alternatives once.

Minor Comments

• Several paragraphs are lengthy and could be split for clarity.

• Table 1 (seven-theme synthesis) could be strengthened to better highlight policy implications.

• Minor typographical and formatting issues should be addressed during revision.

Reviewer #2: Dear author,

I had the pleasure of reviewing your manuscript, which seems well done overall and highlights an issue that still needs to be explored in depth in public health. However, I have a few requests to improve my understanding of the text, as well as that of future readers.

- Please comment and explicit the difference between NPHA and NPHI (if any in the extent of the paper) and clarify in the results whether these are applicable to one or both.

- The case of Ethiopia is well highlighted, but the discussion is more in-depth than for other countries. Please clarify whether there is an objective reason for examining the data relating to Ethiopia in greater detail; otherwise, please reduce the level of detail.

- Introduce practical and operational recommendations in the short term, or at least prioritise the proposed implications.

- The abstract and results refer to the period 2007-2025 (explain the rationale); however, the eligibility criteria refer to 2000-2025. Correct if this is a typo, or explain if there is a real difference in the period.

- In “Strenghts and limitations” there is the following sentence: “Grey literature inclusion helped mitigate this gap but also introduced variability in methodological rigor”. Expand further, considering the potential weight of including grey literature in the review.

**Do you want your identity to be public for this peer review?** For information about this choice, including consent withdrawal, please see our Privacy Policy

Reviewer #1: No

Reviewer #2: **Yes:**  Nunzio Zotti

---

## [Decision Letter · Decision Letter 1]

1 Feb 2026

Building Systems for Preparedness: Global Scoping Studies on Institutional Governance and National Public Health Agencies

PGPH-D-25-03097R1

Dear Dr Sasie,

We are pleased to inform you that your manuscript 'Building Systems for Preparedness: Global Scoping Studies on Institutional Governance and National Public Health Agencies' has been provisionally accepted for publication in PLOS Global Public Health.

Best regards,

Julia Robinson

Executive Editor

Reviewer Comments (if any, and for reference):

Reviewer's Responses to Questions

**Comments to the Author**

Reviewer #1: All comments have been addressed

Reviewer #2: All comments have been addressed

publication criteria?

Reviewer #1: Yes

Reviewer #2: Yes

3. Has the statistical analysis been performed appropriately and rigorously?

Reviewer #1: (No Response)

Reviewer #2: Yes

4. Have the authors made all data underlying the findings in their manuscript fully available (please refer to the Data Availability Statement at the start of the manuscript PDF file)?

Reviewer #1: Yes

Reviewer #2: Yes

5. Is the manuscript presented in an intelligible fashion and written in standard English?

Reviewer #1: Yes

Reviewer #2: Yes

Reviewer #1: I thank the authors for their careful and substantive revision of the manuscript. Overall, the revision meaningfully addresses the major and minor points raised in my initial review, and the manuscript is notably improved in clarity, coherence, and analytical focus.

Major Points

1. Length and redundancy

This concern has been largely and successfully addressed. The Results section is now more clearly focused on empirical patterns and thematic findings, while interpretive and normative commentary has been appropriately shifted to the Discussion. Redundant restatement of findings across sections has been reduced, improving readability and flow. While the manuscript remains comprehensive, the current length is justified by its global scope and synthesis goals.

2. Central analytical thesis

The central analytical argument is now clearly articulated and consistently reinforced. The manuscript explicitly advances the thesis that preparedness effectiveness depends less on the mere existence of institutions than on their integration within legally mandated, autonomous, and sustainably financed governance systems with empowered subnational implementation. This framing is introduced early in the manuscript and revisited in the Discussion, providing a coherent analytical through-line that was less explicit in the original version.

3. Methods section consistency

The authors have successfully harmonized the publication date range, clearly defining the inclusion period as 2000–2025 and providing a transparent rationale for this choice. This resolves the earlier inconsistency between the Abstract and Methods sections and improves methodological clarity for readers.

4. Terminology consistency

Terminology has been substantially improved and standardized. The manuscript now consistently uses public health emergency preparedness and response (PH EPR) as the primary term, with related variants defined once and not used interchangeably thereafter. This change significantly improves conceptual precision and readability.

Minor Points

• Several long paragraphs have been broken up, enhancing clarity without sacrificing content.

• Table 1 has been strengthened and now more clearly highlights policy-relevant implications of the seven-theme synthesis.

• Minor typographical and formatting issues appear to have been corrected.

Overall Assessment

The revision demonstrates strong responsiveness to reviewer feedback and reflects thoughtful engagement with both structural and conceptual critiques. The manuscript is now analytically sharper, methodologically clearer, and better aligned with the expectations of PLOS Global Public Health. I believe it makes a solid contribution to the literature on institutional governance and national public health preparedness and is suitable for publication pending any final editorial adjustments.

Reviewer #2: Dear author, I have reviewed the revised manuscript and believe that my previous comments have been addressed correctly and I do not have any other concern. Best of luck with the submission.

**Do you want your identity to be public for this peer review?** For information about this choice, including consent withdrawal, please see our Privacy Policy

Reviewer #1: **Yes:**  Nikolay Lipskiy, MD, PhD, MBA

Reviewer #2: **Yes:**  Nunzio Zotti
